# Estimating post-operative complication rates in patients with primary brain tumours from routine administrative data: A national cohort study

Radvile Mauricaite[1,2], Alex Bottle[3], Andrew Brodbelt[4], Kerlann Le Calvez[1,2], Peter Treasure[5], Stephen J. Price[6], Thomas C. Booth[7,8], Seema Dadhania[1,2], Jonathan J. Gregory[2], Maureen Dumba[9], Joanne Droney[10,11], Jawad Basharat[12], Matt Williams[1,2]*

1 Department of Radiotherapy, Charing Cross Hospital, Imperial College Healthcare NHS Trust, London, United Kingdom, 2 Computational Oncology Laboratory, Institute of Global Health Innovation, Imperial College London, London, United Kingdom, 3 School of Public Health, Imperial College London, London, United Kingdom, 4 The Walton Centre NHS Foundation Trust, Lower Lane Liverpool, United Kingdom, 5 Peter Treasure Statistical Services Ltd, King's Lynn, United Kingdom, 6 Division of Neurosurgery, Department of Clinical Neurosciences, University of Cambridge, Cambridge, United Kingdom, 7 Department of Neuroradiology, Ruskin Wing, King's College Hospital NHS Foundation Trust, London, United Kingdom, 8 School of Biomedical Engineering & Imaging Sciences, King's College London, London, United Kingdom, 9 Department of Neuroradiology, National Hospital for Neurology and Neurosurgery, University College London NHS Foundation Trust, London, United Kingdom, 10 Symptom Control and Palliative Care Team, The Royal Marsden NHS Foundation Trust, London, United Kingdom, 11 Department of Surgery & Cancer (Honorary Clinical Senior Lecturer), Imperial College London, London, United Kingdom, 12 Department of Clinical Coding, Imperial College Healthcare NHS Trust, London, United Kingdom

* matthew.williams@imperial.ac.uk

## Abstract

### Introduction

Neurosurgery is an important element of brain tumour treatment but carries with it the risk of complications. Previous work has defined a narrow set of general post-operative complications which are used as Patient Safety Indicators (PSIs), but these are not brain tumour specific and do not capture the full range of complications. As a result, there is no way of measuring post-operative complications in neurosurgery at a national level.

### Methods

We conducted a retrospective, observational cohort study using a comprehensive national administrative dataset from England on adult brain or spinal tumour patients to better define post-operative complications. We generated and validated a new list of post-operative complications – ICL list. The ICL list contains codes selected specifically from our Gliocova dataset combined with general OECD-defined PSI list. The ICL list is novel as it can assess brain tumour patient complications using an

**Data availability statement:** We are open to collaborating with research teams as we believe in sharing knowledge. If people are interested in working on Gliocova, they can contact us to enquire about data access. The data is securely held on Imperial College London servers so arrangements will need to be made prior to access. We have a data-sharing agreement with NHS England/Digital that prevents us from free-ly sharing data, as we have access to individual patient level NHS data. Furthermore, we have REC approvals (REC reference: 16/YH/0213) that only allow us to share aggregated data and not pseudo-anonymised data. You can contact the ethics committee here: sheffield.rec@hra.nhs.uk.

**Funding:** This work was funded by National Institute for Health Research (NIHR) Biomedical Research Council (BRC), RM Partners hosted by the Royal Marsden NHS Foundation Trust and Imperial College Healthcare NHS Trust. SJP is funded by a NIHR Career Development Fellowship (CDF-18-11-ST2-003). The views expressed are those of the author(s) and not necessarily those of the NIHR or the Department of Health and Social Care. ABo is funded by the National Institute for Health and Care Research. TCB is funded by the Wellcome Trust 2022-2023 (Award number: 203148/A/16/Z).

**Competing interests:** Dr. Williams is employed by Imperial College Healthcare NHS Trust. He is also the medical director of PearBio (salary and share options). Stephen J. Price is an advisor for TUMOURVUE Ltd (no financial involvement) and is the Chair of the Education Committee of the European Association for Neuro-oncology (EANO). He is on the speaker board for Medac GmbH and organises courses to train surgeons to use 5-ALA for which he is reimbursed, but has not run such a course since 2019. Alex Bottle declared obtaining fees from AstraZeneca and Lilly outside the submitted work. This does not alter our adherence to PLOS ONE policies on sharing data and materials.

administrative dataset at a national level and captures more specific brain tumour related complications.

## Results

In our study, 30-day readmission after surgery was 12.7% and 30-day mortality was 2.3%. The ICL list of complications identified many more patients with complications (N = 3,274 (11.3%)) compared to OECD-defined PSI list (N = 568 (2.0%)) without reducing model performance. 30-day mortality was 6.5% in those with complications and 1.8% in those without.

## Discussion

We have identified a much wider set of complications than the OECD-defined PSIs and shown that patients developing these have worse outcomes than those without complications. This enables us to estimate the risk of post-operative complications in brain tumour patients using national administrative data. It forms the basis for planned further work, allowing us to explore the predictors of and consequences of post-operative complications.

## Introduction

Primary Brain and Central Nervous System (CNS) tumours are a group of rare, heterogenous diseases, including benign and malignant tumours, most of which occur in the brain rather than the spine. The commonest tumours are Glioblastoma (GBM; WHO Grade 4 glioma) and meningioma, which between them account for >50% of all adult brain tumours. GBM is an aggressive, malignant tumour, whereas meningiomas are largely benign. The treatment pathways and outcomes are thus very different.

Neurosurgery is a key treatment in brain tumour patients, but it is not without risk. Neurosurgery is very centralised in England, and England has a robust national cancer data infrastructure, which holds data on pathology, treatment, admissions and survival [1]. Post-surgical death is an obvious post-operative complication, but since mortality rates are low, it is not a sensitive indicator of quality [2]. Both the Organisation for Economic Co-operations and Development (OECD) and the US Agency for Healthcare Research and Quality (AHRQ) have developed Patient Safety Indicators (PSI), which define a small number of key diagnoses (e.g., post-operative Pulmonary Embolus) [3,4]. However, these are a very restricted set of complications, meant to be used as quality indicators across wide range of surgical practice, and to allow for trans-national comparisons of patient safety, rather than as a measure of the rate of complications. In addition, not all AHRQ PSI can be translated to the ICD-10-WHO system used in the UK [5], and the OECD-defined PSI list does not have any CNS specific indicators [6]. Research on patient safety and complications in brain tumour patients is limited, and most of it comes from the US [7–10]. Previous work from Europe and the UK includes the development and validation of a Post-operative

Morbidity Survey for patients undergoing cranial neurosurgery [11] and complications after neurosurgery using either Landriel or Clavien-Dindo classifications [12,13]. However, these require review of clinical notes and patient examination or questionnaires, which limits the feasibility of conducting such work at scale. As a consequence, we currently have no way of measuring complications (outside of a very narrow set of PSI) at scale at a national level. This makes it impossible, for example, to even count the number of patients who have post-operative complications.

Large administrative datasets have been used to assess outcomes after procedures such as hip replacement and cardiac surgery [14]. Such datasets allow large scale assessment of outcomes and co-morbidities [15,16]. However, assessing the validity of the new definition of complications is challenging: by definition, we cannot conduct patient-level notes review. We are engaged in a trade-off: counting every post-operative diagnosis as a complication will capture too many patients; counting only the PSI indicators captures too few. One approach is examine plausible downstream effects of complications (such as mortality and length of stay). If the performance of statistical models stays the same as we expand the list of complications, then it suggests that the expanded list is valid. In this work, we use data from the GlioCova project to define a new set of post-operative complications (ICL list) and examine the relationship between our novel definitions of complications and 30-day mortality and length of stay in adult patients undergoing first surgical intervention for a primary CNS tumour [17]. Our key aim is to be able to estimate the number of complications; we validate this by assessing whether models constructed using the newly defined ICL set of complications have similar performance to models constructed using existing (PSI) definitions of complications. We use the ICL set of complications to estimate, for the first time, rates of complications in a large, complete, multi-year national cohort of primary brain tumour patients undergoing neurosurgery in England.

## Methods

### Data

The GlioCova project holds data on all adults with brain tumours diagnosed in England from 2013–2018, with treatment and survival data available until August 2020. Data was extracted on the 10th of August 2020, with mortality data (= survival) censored in October 2022. It contains linked pseudo-anonymised data on patients' demographics, surgery, chemotherapy, radiotherapy, hospital admissions, imaging and death. Authors had no access to information that could identify individual participants during or after data collection. Data also includes deprivation status based on the Index of Multiple Deprivation (IMD). This is calculated based on postcode as a marker of socio-economic status at a small local area level across England and grouped into quintiles with IMD quintile 1 representing areas with lowest deprivation and quintile 5 areas with most deprivation [18]. Analysis is supported by the Expert Advisory Group (EAG), a multi-disciplinary group of data analysts, clinicians, patients, carers and charities. More details on ethical approval and coding systems used can be accessed in S1 Appendix.

### Patient cohort

We identified all patients with a primary brain tumour who underwent surgical resection or biopsy ("surgical intervention") (see S2 Table), and we excluded those who had a diagnosis of another cancer. Patients who were recorded as having a surgical resection and biopsy on the same day were considered to have a surgical resection. We only considered the first admission for surgical intervention for each patient (index admission). We excluded patients who had a clearly erroneous (> −5000 days) interval from diagnosis to procedure and removed duplicated admissions in 39 patients, retaining the admission with the shortest length of stay (LOS). The remaining patients were our analytical cohort. Although we considered the whole cohort, we explored rates of outcomes in patients with meningioma and glioblastoma as clinical exemplars.

### Defining the ICL post-surgical complication list

We used the list of diagnosis codes from the 2018 OECD Patient Safety Questionnaire [6], excluding those related to obstetric trauma (see S3 Table). This is the OECD-defined PSI list. We separately identified all diagnoses from the GlioCova

dataset during the first intervention, excluded brain/spine tumour diagnoses and those relating to chronic illnesses (e.g., hypertension, asthma), and selected the 100 most common remaining diagnoses after these exclusions (see S4 Table).

We used a modified Delphi [19] approach starting with the 100 most common diagnoses and three review rounds to refine the set of potential post-surgical complications as defined below (see S5 Fig). Diagnoses were reviewed by a group of five experienced attending clinicians (of whom at least 4 have more than 5-years of experience as attending consultants) who reviewed the diagnoses and selected potential complications. They were asked to select potential post-surgical complications and these were defined as `events and complications of care – even if recognised as being known complications'. As it is not clear whether a patient had a diagnosis code before admission or developed it afterwards, we asked to select only those codes that are definitely indicating a post-surgical complication by asking the following: 'would you usually think this sign or symptom was a complication?'. The interrater variability for the first round was acceptable with Krippendorff's Alpha = 0.7484, Fleiss' Kappa = 0.7480 and pairwise agreement percentages varying from 0.786 to 0.954.

In the second round experts (4 clinicians and one clinical coder) voted on the codes that were not unanimously selected in the first round (selecting by one or more). The interrater agreement was very low indicating disagreement between voters with Krippendorff's Alpha = −0.077, Fleiss' Kappa = −0.084 and pairwise agreement percentage varying from 0.323 (poor agreement) to 0.645 (fair to good agreement). From this round that evaluated codes that did not have a unanimous agreement in the first round, we took all the codes selected either by the specialist clinical coder or by at least three out of four clinicians.

We then reviewed suggested diagnoses against a local dataset where we able to conduct manual notes review of 327 patients who had undergone neurosurgery, of whom 37 had both a brain tumour and a possible complication. Based on this, we retained only 11 (27.5%) out of 41 previously selected codes that were definitely post-operative complications. Finally, we combined the diagnosis codes selected from the GlioCova dataset with those used in the OECD Patient Safety Questionnaire into our new list of complications – ICL list. We then validated the ICL list by constructing statistical models using both the OECD-defined PSI list and ICL list of complications, and comparing performance of the models using OECD-defined complications and ICL complications.

### Assessing co-morbidities

We measured co-morbidities using a modified Elixhauser index [15], removing brain/spinal tumour diagnoses, adding dementia [16] and derived weights using all admissions in our dataset and previously defined methods [20] (see S6 and S7 Tables).

### Outcome measures

Our outcome measures were 30-day mortality rate and length of stay (in days). We defined 30-day mortality as those who had died within 30 days of first surgical intervention. LOS was defined as the interval in days between admission and discharge. We have defragmented admissions and combined LOS if patients were discharged to another hospital. We have removed patients that did not have a LOS available. In order to simplify model presentation, we dichotomised LOS by considering LOS of <= 3 days vs > 3 days based on data distribution as there was the highest number of patients that had a LOS of 3 days (N = 3,703). We have removed patients with very long LOS (>180 days) for LOS analysis as these were outliers. Since GBM and meningioma are the commonest primary brain tumours and have different rates of complications, we calculated rates of complications and LOS in patients with GBM vs. meningioma, and by broad age bands to illustrate the impact of these on complication rates and LOS.

### Statistical approaches

To evaluate how the OECD-defined PSI list and ICL complication list were associated with 30-day mortality and LOS, logistic regression was performed in R (v. 4.1.2) using glm function [21,22]. Complications were defined as above, but for analysis of LOS we did not consider patients who were discharged from the neurosurgical centre to

another hospital as this would complicate the statistical modelling. Both unadjusted and adjusted regression models were developed. Adjusted regression models also included other predictor variables such as deprivation status, tumour/intervention type, co-morbidities. Variable selection in the adjusted analyses was performed using backward elimination based on the Akaike information criterion (AIC). Age and sex were included in the models even if eliminated during AIC elimination due to their a priori importance. If there were differences in the models using OECD-defined PSI list compared to ICL list, we used all of the variables that were selected as significant in either of the two models. Discrimination was assessed by inspection of ROC curves and area under the curve (C-statistic). DeLong test was performed to calculate the p-value for the difference in the AUC of the ROC curves [23]. We compared the performance of models developed from the OECD-defined PSI list and the ICL list by comparing the AUC between the two models. Please note, the models were only used to validate the ICL list and not to predict outcomes.

### Ethics statement

This data uses anonymised pooled data from multiple national datasets. In line with published guidance, such work does not require individual informed patient consent. We have research ethics (Yorkshire and the Humber – Sheffield Research Ethics Committee) approval for this work. All published data are anonymised and non-identifiable.

There is a data-sharing agreement (DSA) in place between NHS England/Digital and our team ("Computational Oncology Laboratory" hosted at Imperial College London) that specifies that we're allowed to publish aggregated patients' data in peer-reviewed journals. Further to the DSA, we also have Research Ethics Committee (REC) approvals (REC reference: 16/YH/0213) that gave us permission to publish aggregated patients' data in peer-reviewed journals.

## Results

### Patient cohort and outcomes

We identified 29,467 patients with a primary CNS tumour who underwent a surgical intervention. We excluded 318 patients who underwent biopsy and surgery on different days (see S8 Fig), 81 patients with an incorrect interval from diagnosis to operation, and a further 7 patients who were duplicates. We further removed 18 patients that did not have an interval from diagnosis to vital status, 23 patients who had left the UK (and therefore vital status was unknown) and 2 patients that had incorrect intervals from diagnosis to vital status or surgical intervention (died before surgery date). Thus, the final analytical cohort consisted of 29,018 patients. The median age was 58 (Q1 = 18, Q2 = 46, Q3 = 58, Q4 = 68, Q5 = 94) and 50.5% were male. Patient characteristics can be found in Table 1. Most patients were discharged to 'usual place of residence' (8% to another hospital; 1% to care home or a hospice) (see S9 Table). The most common reasons for readmission were headache, further surgery, post-operative infection, CSF leak, and nausea and vomiting (see S10 Table).

### Defining complications

The newly defined ICL list of complication codes consisted of 51 post-surgical complications (40 OECD-defined and 11 Gliocova-defined (see Table 2). Most patients who had a complication had one from the Gliocova-defined list (90.3%, n = 2,956) and a third of the 568 patients who had an OECD-defined complication also had a Gliocova-defined complication (44.0%, n = 250). See Fig 1 for the selection of complication and no-complication patient cohorts. Patients with complications from our ICL list had much higher rates of 30-day mortality compared to patients without complications – 6.5% (212 out of 3,274 patients) for patients with complications compared to 1.8% (465 out of 25,744 patients) in the no-complications group. They also had longer LOS (median LOS was 17 days compared to 5 days). Patients with

**Table 1. Patient characteristics.**

| Characteristics | Value |
|---|---|
| Total patients | 29,018 |
| Gender | |
| *Male* | 14, 650 (50.5%) |
| *Female* | 14, 368 (49.5%) |
| Age (in years) | |
| *Median* | 58 |
| *Q1* | 18 |
| *Q2* | 46 |
| *Q3* | 58 |
| *Q4* | 68 |
| *Q5* | 94 |
| Tumour diagnosis *(ICD-10 + ICD-O-2 codes)* | |
| *C71 Glioblastoma* (C71 + 9440/9441/9442) | 11,098 (38.2%) |
| *C71 Other* (C71 + any other morphology code): | 4,830 (16.5%): |
| Astrocytoma (9400) | 1,108 (3.8%) |
| Anaplastic astrocytoma (9401) | 991 (3.4%) |
| Oligodendroglioma (9450) | 680 (2.3%) |
| Anaplastic oligodendroglioma (9451) | 513 (1.8%) |
| Glioma, malignant (9380) | 419 (1.4%) |
| Pilocytic astrocytoma (9420) | 260 (0.9%) |
| Anaplastic oligoastrocytoma (9382) | 186 (0.6%) |
| Ependymoma (9391) | 146 (0.5%) |
| All other morphologies | 527 (1.8%) |
| *Meningioma* (D32.0 + 9530) | 3,894 (13.4%) |
| *Vestibular Schwannoma* (D33.3) | 1,326 (4.5%) |
| *Other* | 7,870 (27.1%) |
| Intervention type | |
| *Resection* | 27,817 (95.9%) |
| *Biopsy* | 1,201 (4.1%) |
| Deprivation status | |
| *1 – Least deprived* | 6,545 (22.6%) |
| *2* | 6,470 (22.3%) |
| *3* | 5,943 (20.5%) |
| *4* | 5,208 (18.0%) |
| *5 – Most deprived* | 4,852 (16.7%) |

complications from the OECD-defined PSI list only also had higher 30-day mortality (7.6%) and LOS (median LOS was 20 days). See Table 3.

## Model performance using ICL list versus OECD only

Intervention type, ethnicity, age, comorbidities (ECI score), tumour type and income quintile were identified as significant predictors of 30DM and in addition sex was identified as significant predictor of LOS. In both adjusted and unadjusted analyses, regression models for 30-day mortality and longer LOS using the ICL complication list performed similarly or better than models using the OECD-defined PSI list only (Figs 2 and 3). In unadjusted analysis, ROC curves were slightly

**Table 2. Full ICL complication list.**

| ICD-10 code | Diagnosis description |
| --- | --- |
| **OECD Patient safety (PS) questionnaire codes** | |
| T81.5 | Foreign body accidentally left in body cavity or operation wound following a procedure |
| T81.6 | Acute reaction to foreign substance accidentally left during a procedure |
| Y61.0 | Foreign object accidentally left in body during surgical and medical care: During surgical operation |
| Y61.1 | Foreign object accidentally left in body during surgical and medical care: During infusion or transfusion |
| Y61.2 | Foreign object accidentally left in body during surgical and medical care: During kidney dialysis or other perfusion |
| Y61.3 | Foreign object accidentally left in body during surgical and medical care: During injection or immunization |
| Y61.4 | Foreign object accidentally left in body during surgical and medical care: During endoscopic examination |
| Y61.5 | Foreign object accidentally left in body during surgical and medical care: During heart catheterization |
| Y61.6 | Foreign object accidentally left in body during surgical and medical care: During aspiration, puncture and other catheterization |
| Y61.7 | Foreign object accidentally left in body during surgical and medical care: During removal of catheter or packing |
| Y61.8 | Foreign object accidentally left in body during surgical and medical care: During other surgical and medical care |
| Y61.9 | Foreign object accidentally left in body during surgical and medical care: During unspecified surgical and medical care |
| I26.0 | Pulmonary embolism with mention of acute cor pulmonale |
| I26.9 | Pulmonary embolism without mention of acute cor pulmonale |
| I80.1 | Phlebitis and thrombophlebitis of femoral vein |
| I80.2 | Phlebitis and thrombophlebitis of other deep vessels of lower extremities |
| I80.3 | Phlebitis and thrombophlebitis of lower extremities, unspecified |
| I80.8 | Phlebitis and thrombophlebitis of other sites |
| I80.9 | Phlebitis and thrombophlebitis of unspecified site |
| I82.8 | Embolism and thrombosis of other specified veins |
| A40.0 | Septicaemia due to streptococcus, group a |
| A40.1 | Septicaemia due to streptococcus, group b |
| A40.2 | Septicaemia due to streptococcus, group d |
| A40.3 | Septicaemia due to streptococcus pneumoniae |
| A40.8 | Other streptococcal septicaemia |
| A40.9 | Streptococcal septicaemia, unspecified |
| A41.0 | Septicaemia due to staphylococcus aureus |
| A41.1 | Septicaemia due to other specified staphylococcus |
| A41.2 | Septicaemia due to unspecified staphylococcus |
| A41.3 | Septicaemia due to haemophilus influenza |
| A41.4 | Septicaemia due to anaerobes |
| A41.5 | Septicaemia due to other gram-negative organisms |
| A41.8 | Other specified septicaemia |
| A41.9 | Septicaemia, unspecified |
| R57.2 | Septic shock |
| R57.8 | Other shock |

*(Continued)*

**Table 2.** (Continued)

| ICD-10 code | Diagnosis description |
|---|---|
| R65.0 | Systemic Inflammatory Response Syndrome of infectious origin without organ failure |
| R65.1 | Systemic Inflammatory Response Syndrome of infectious origin with organ failure |
| T81.1 | Shock during or resulting from a procedure, not elsewhere classified |
| T81.3 | Disruption of a wound not elsewhere classified |
| **Additional Gliocova-defined codes** | |
| T81.0 | Haemorrhage and haematoma complicating a procedure, not elsewhere classified (at any site resulting from a procedure) |
| N39.0 | Urinary tract infection, site not specified |
| G96.0 | Cerebrospinal fluid leak |
| J18.1 | Lobar pneumonia, unspecified |
| G97.8 | Other postprocedural disorders of nervous system |
| I63.9 | Cerebral infarction, unspecified |
| B96.2 | Escherichia coli [E. coli] as the cause of diseases classified to other chapters |
| J22.X | Unspecified acute lower respiratory infection |
| Y60.0 | Unintentional cut, puncture, perforation or haemorrhage during surgical and medical care: during surgical operation |
| T81.4 | Infection following a procedure, not elsewhere classified* |
| T81.2 | Accidental puncture or laceration during procedure |

better for models using the ICL complication list. AUC for 30-day mortality model using the ICL list was 0.60 (95% CI 0.59–0.62) compared to 0.52 (95% CI 0.51–0.53) with a p-value = 2.7e$^{-15}$ for the difference between the two AUCs. AUC in LOS model was 0.55 (95% CI 0.55–0.55) vs 0.51 (95% CI 0.51–0.51)) accordingly with a p-value = 2.2e$^{-16}$. In the adjusted models the C-statistic (AUC) for the 30-day mortality model using ICL complications was 0.77 (95% CI 0.76–0.79) vs. 0.74 (95% CI 0.73–0.76) for the OECD-model with a p-value = 0.009, and for LOS it was 0.66 (95% CI 0.65–0.67) vs. 0.64 (0.63–0.65) accordingly with a p-value = 7.8e$^{-05}$.

## Rates of outcomes by clinical group

30-day mortality was 2.3 (n = 677) and 12.7% (n = 3,701) of patients were re-admitted within 30 days. Using the ICL list of complications, the overall risk of complication was 11.3% (N = 3,274) (compared to 2.0% (N = 568) using the OECD-defined PSI list). The complications from the ICL list that were driving the increase in complication rate most were 'Haemorrhage and hematoma complicating a procedure, not elsewhere classified' (N = 953, 29%), 'Urinary tract infection, site not specified' (N = 673, 20.6%), 'Lobar pneumonia, unspecified' (N = 260, 7.9%), 'Other postprocedural disorders of nervous system' (N = 227, 6.9%) and 'Cerebral infarction, unspecified' (N = 211, 6.5%). Patients with glioblastoma had a higher risk of 30-day mortality (4.2% compared to 1.2%) and 30-day readmission (14.6% compared to 11.0%) compared to meningioma patients and the overall rates of these outcomes across all age categories (30-day mortality was 5.1%, 3.7% and 4.6% for the 18–40, 40–65 and the 65 + age categories accordingly compared to 0.5%, 0.6% and 2.4% and 30-day readmission was 17.5%, 14.5% and 14.4% compared to 11.1%, 11.3% and 10.4%). Nonetheless, risk of having post-surgical complications was higher in the meningioma patient group (13.6%, 12.0% and 20.0% for meningioma patients for the 3 age categories compared to 8.9%, 7.2%, 9.9% for the glioblastoma patients). See Table 4.

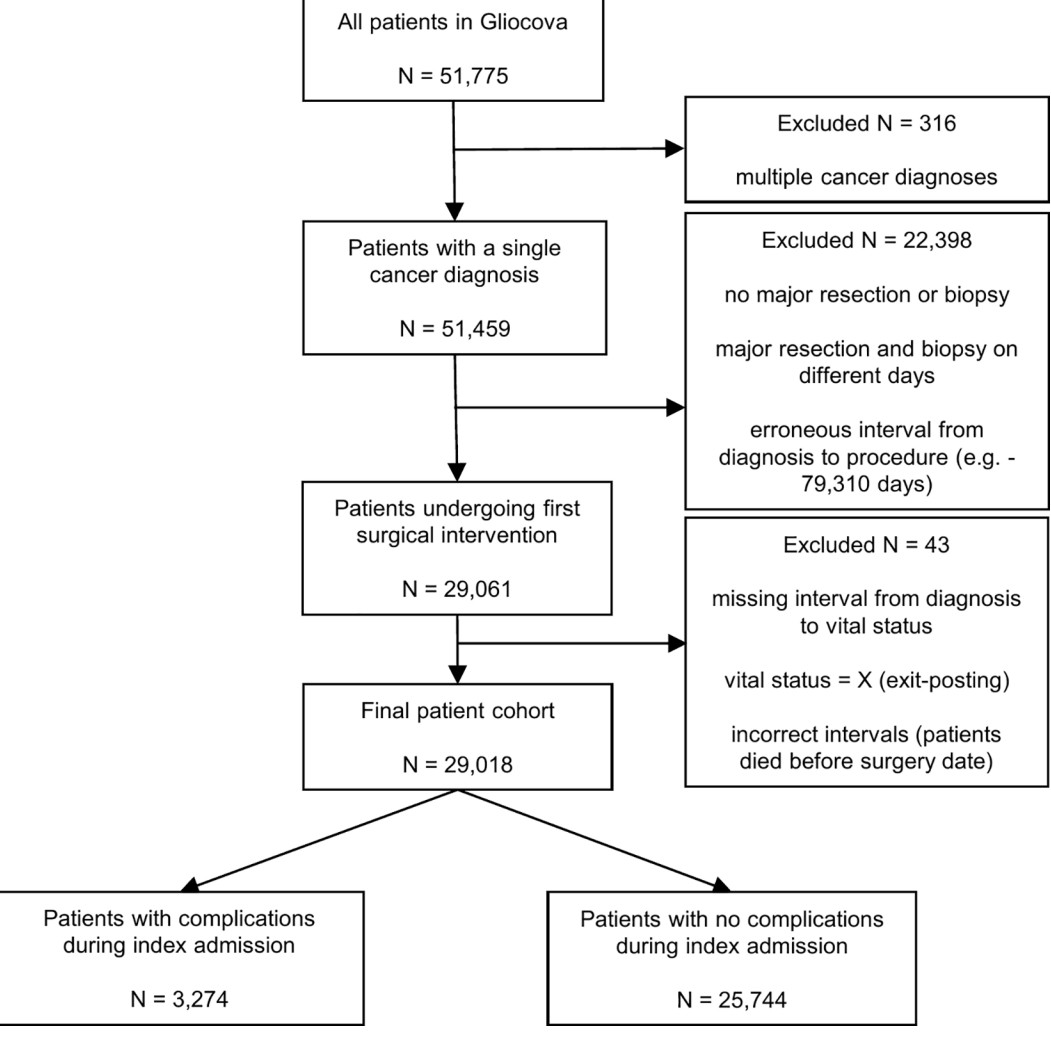

**Fig 1. Flowchart for the selection of patients with and without complications from the patient cohort.**

**Table 3. Outcomes (30-day mortality, median LOS) for patient subgroups based on complication status (ICL defined complications versus OECD-defined PSI list versus No complications).**

| | Complication status | | |
| --- | --- | --- | --- |
| | **ICL defined complications** | **OECD-defined PSI list** | **No complications** |
| **30-day mortality** | 6.5% | 7.6% | 1.8% |
| **Median LOS** | 17 days (Q1 = 0, Q2 = 8, Q3 = 17, Q4 = 35, Q5 = 178) | 20 days (Q1 = 0, Q2 = 10, Q3 = 20, Q4 = 36, Q5 = 178) | 5 days (Q1 = 0, Q2 = 3, Q3 = 5, Q4 = 10, Q5 = 169) |

## Discussion

We have used a large, comprehensive cohort of all adults undergoing first surgical intervention for a primary brain tumour over a six-year period in England to define an expanded set of post-operative complications. We have shown the OECD-defined PSI capture a small proportion of patients with complications compared to the newly defined ICL

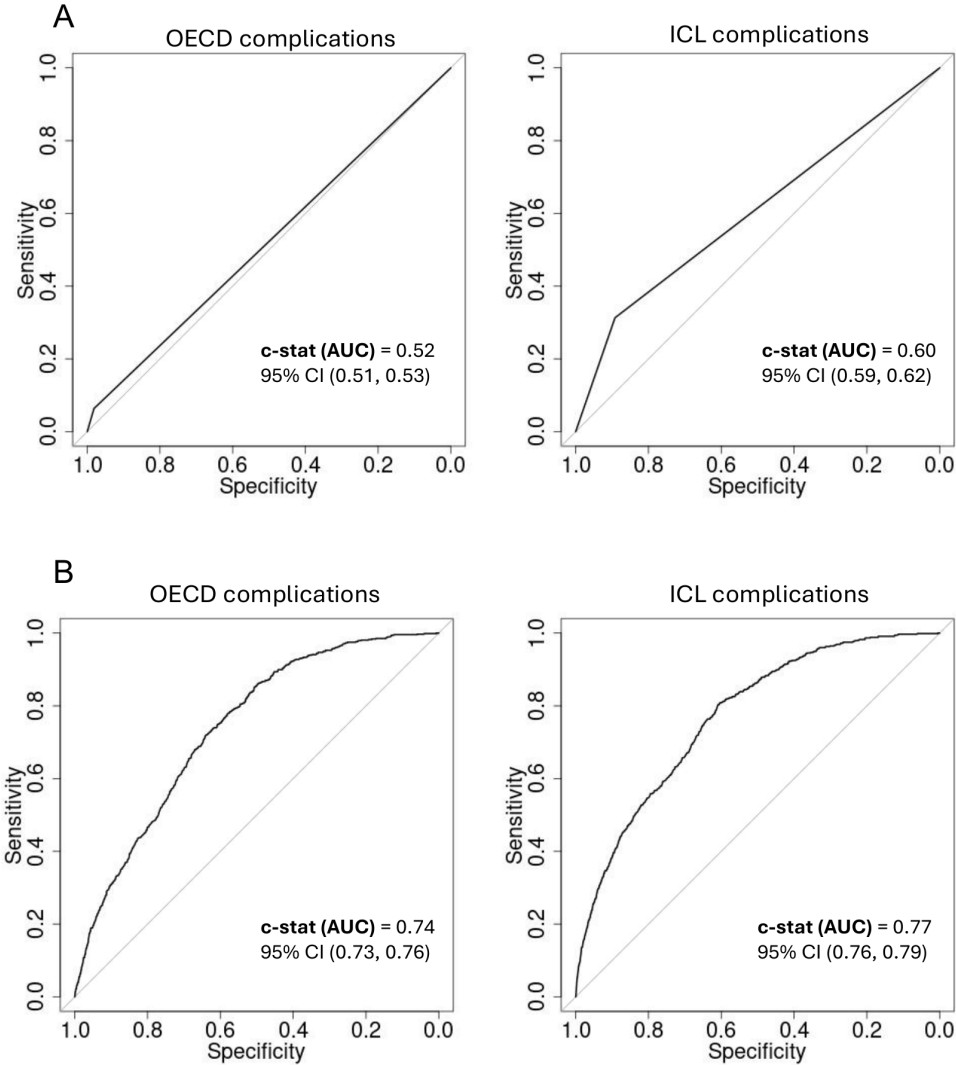

**Fig 2. ROC curves and the c-statistic for unadjusted and adjusted models predicting 30-day mortality using ICL complication list versus OECD-defined PSI list only. (A)** ROC curves of unadjusted models. **(B)** ROC curves of the adjusted models. Adjusted models included intervention type, ethnicity, age, comorbidities (ECI score), tumour type and complications variables.

list of complications. However, models developed using the ICL list of complications perform as well as those using the OECD-defined PSI list in predicting 30-day mortality and length of stay. Unlike previous approaches, this assessment of complications can be done entirely using routine healthcare data, rather than manually curated data. As a consequence, we were able to analyse a far larger dataset than previous work.

30-day mortality rates were low (2.3%), the complication rate was 11.3%, and approximately 1 in 8 patients were readmitted within 30 days of surgery. Patients who were older or had more co-morbidities, were more likely to have complications and more likely to die. While not surprising, over 40% of patients had at least one co-morbidity, and the median age was 58, with 18% being over 70. The need to understand neurosurgical outcomes in an older, co-morbid population is thus clear. Complication rates for patients with meningioma were higher than those for GBM; we think that this is due to clinical decision making around risk, given that surgery may be curative for patients with

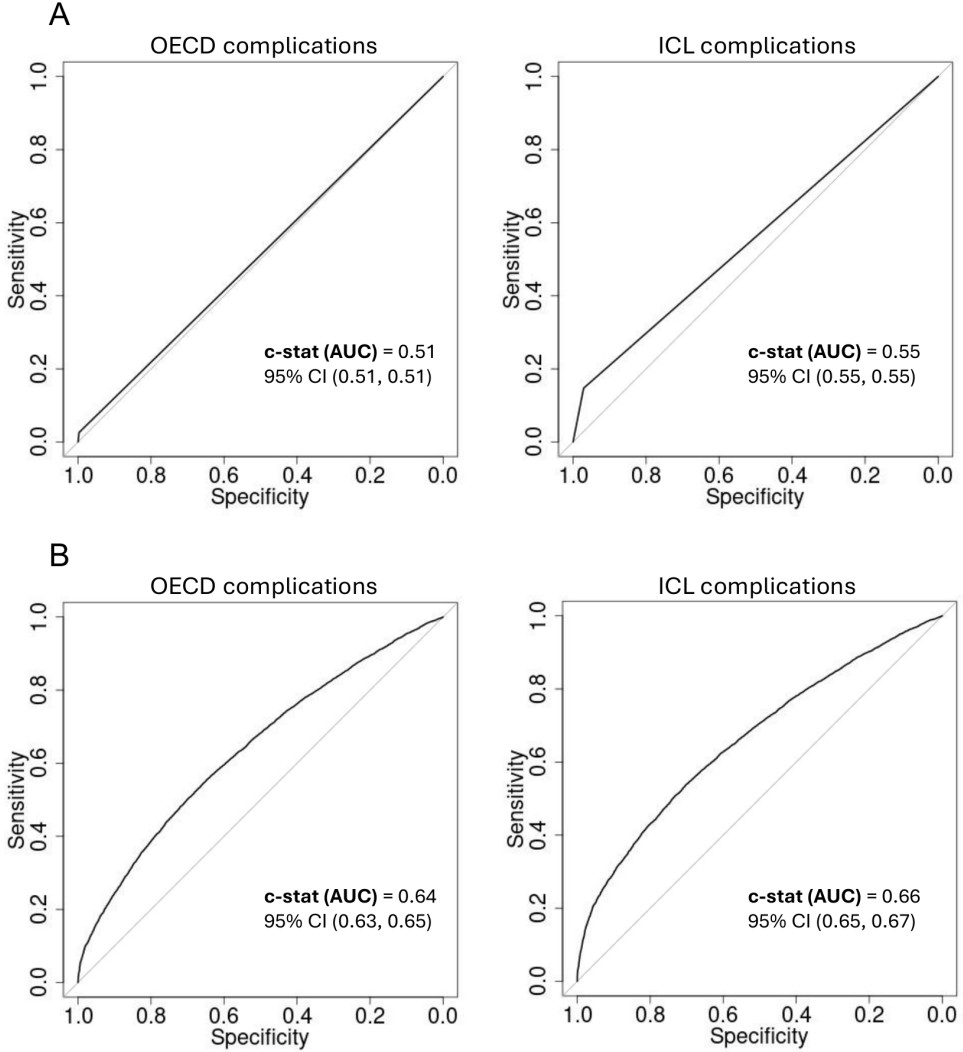

**Fig 3. ROC curves and the c-statistic for unadjusted and adjusted models predicting longer length of stay using ICL complication list versus OECD-defined PSI list only. (A)** ROC curves of unadjusted models. **(B)** ROC curves of the adjusted models. Adjusted models included intervention type, ethnicity, age, comorbidities (ECI score), tumour type, sex, income quintile and complications variables.

meningioma, and thus both patients and clinicians are willing to accept a higher risk of complications. However, we cannot prove this without detailed review of notes and interviews with clinicians and patients, which is not feasible in this type of work.

Our study is considerably larger than most previous work and post-operative outcome rates are similar to those found in previous research. Our reported 30-day mortality of 2.3% (4.2% in glioblastoma patients and 1.2% in meningioma patients) is similar to 3.3% after craniotomy for primary malignant brain tumours in a national US dataset, 3.6% in a small intra-axial brain tumour patient surgery cohort from India, 5.2% in a Dutch study of first-time glioblastoma surgery and 3% in our previous work using a slightly different time period and patient cohort [2,24–26]. The 30-day readmission rate of 12.7% (14.6% in glioblastoma patients and 11.0% in meningioma patients) is also similar to previous work: 11.5% after craniotomy for primary malignant brain tumours in a US study using National Surgical Quality Improvement Program

**Table 4. Rates of outcomes (30-day mortality, complications, 30-day readmission, median LOS) for patient subgroups based on age and tumour type.**

| | Age group | | |
| --- | --- | --- | --- |
| | **18-40** | **40-65** | **65+** |
| **Glioblastoma** | 5.1%, 8.9%, 17.5%, 5 days (IQR=7) | 3.7%, 7.2%, 14.5%, 5 days (IQR=6) | 4.6%, 9.9%, 14.4%, 5 days (IQR=9) |
| **Meningioma** | 0.5%, 13.6%, 11.1%, 6 days (IQR=7) | 0.6%, 12.0%, 11.3%, 6 days (IQR=6) | 2.4%, 20.0%, 10.4%, 8 days (IQR=14) |
| **All** | 1.4%, 10.5%, 13.4%, 5 days (IQR=5) | 2.1%, 10.2%, 12.7%, 5 days (IQR=6) | 3.5%, 13.3%, 12.4%, 6 days (IQR=10) |

registry data (2005–2015), 13.2% readmission incidence after craniotomy for malignant supratentorial tumours in California (1995–2010), and 17.8% after craniotomy in brain metastasis patients in a US study [24,27,28].

Comparing post-surgical complication rates is challenging due to the varying sizes of datasets and different definitions used to define post-operative complications, and a review of methodological issues is given by Keltie et al. [29]. Nonetheless, the complication rate using our approach (11.3%) is similar to previous literature, but dependent on the definitions used. For example, one previous study quoted a very small complication rate of 3.4%, but this is due to them using a very short list of defined complications (retention of a foreign object, wrong side surgery, iatrogenic stroke, meningitis or haemorrhage) [30]. A French study had a 31% complication rate (25% with post-operative nausea and vomiting (PONV), 16% with neurologic complications) but was very small (N=167) and only covered one hospital [31]. Another French study identified severe complications, which occurred in around 11% of patients [32]. Some large american studies that have used complications or adverse events as defined by the American College of Surgeons National Surgical Quality Improvement Program (NSQIP) recorded complication rates of 8.2%, 14.3% and 25% in neurosurgery [33–35]. Some other studies investigating neurosurgical procedures had complication rates varying from 7.8% to 25% [36–38]. See Table 5 for comparison below.

This is the first study to take a comprehensive national view of complications after brain tumour surgery. However, the use of administrative data does not allow us to provide detailed clinical information (e.g., on severity of complications) and some diagnoses might be miscoded or missed [39–42]. There may be an underlying tendency to capture all complications (as it may increase the income for the hospital for that stay); equally, we know that many diagnoses are missed from routine electronic healthcare records, and while there is a risk of bias, it is not clear in which direction it might move our estimate. Equally, we have not conducted a formal sensitivity analysis, although 29% of the complications are due to haemorrhage and 20% due to urinary tract infections, and thus are likely to be robust. However, using administrative data does allow us to systematically explore predictors of outcomes and report both absolute and relative risks in a large, unselected cohort. The combination of data, analysts and our Expert Advisory Group has also allowed us to help refine our questions and analysis. Unlike much previous work, we start with an incident national cohort and have comprehensive tracking of admissions and death. In addition, the Delphi review process was difficult to conduct robustly, due to the distributed nature of the advisory group and their time commitments, as well as the limitations imposed by the COVID-19 pandemic.

The OECD-defined risk of complications is specific – it identifies a small group of patients who have a markedly increased risk of death (7.6% 30-day mortality compared to 6.5% using the ICL complications list and 2.2% in the no-complications group) – but not sensitive, as it misses a large number of patients who have post-operative complication. The vast majority of those who we found to have a post-operative complication were not identified by the OECD-defined PSI list, and the group defined using the combined set of complications had a risk of 30-day mortality that was still 5 times higher than the no-complications group.

**Table 5. Complication rates in our study and in other studies.**

| Study (year) | Population | Type of complication | Complication rate with 95% confidence intervals if available |
|---|---|---|---|
| Our current study (2025) | 29,018 | Newly defined ICL list of complications | **11.3%** (95% CI [10.92, 11.56]) |
| Boissonneau et al. (2023) [36] | 953 | Postoperative infection and other | **21.6%** |
| Meyer et al. (2022) [35] | 4,176 | Adverse events classified according to the American College of Surgeons National Surgical Quality Improvement Program 30-day outcome complication definition (ACS-NSQIP) | **25%** |
| Kulikov et al. (2022) [37] | 514 | Infection (wound, pulmonary, blood stream, urinary tract infection, or central nervous system infection) within first post-operative week | **7.8%** |
| Sahin et al. (2021) [38] | 136 | Thrombotic complications, Hemorrhagic complications, Pulmonary complications, cardiac complications, neurological complications, local complications, infection | **10.3%** |
| Cinotti et al. (2018) [32] | 1,094 | Postoperative neurologic complications requiring in-intensive care unit management | **10.8%** |
| Lonjaret et al. (2017) [31] | 188 | Postoperative nausea and vomiting (PONV), neurologic complications) | **31%** |
| Cote et al. (2016) [33] | 94,621 | The American College of Surgeons National Surgical Quality Improvement Program (NSQIP) complications | **8.2%** (5.6% for spinal patients, 16.1% for cranial patients) |
| De La Garza-Ramos et al. (2016) [30] | 16,530 | Wrong side surgery, retention of a foreign object, iatrogenic stroke, meningitis, hemorrhage/hematoma complicating a procedure, and neurological complications | **3.4%** |
| Rolston et al. (2014) [34] | 38,396 | Complications documented in the NSQIP database | **14.3%** (11.2% for spinal procedures and 23.6% for cranial procedures) |

Despite the comprehensive multi-year cohort, the number of patients undergoing a biopsy is still relatively small; it is therefore not possible to factors specific to biopsy patients vs. Patients undergoing resection. The C-statistic for all models is relatively low (below 0.8), suggesting that there are significant other factors that influence outcomes. It is therefore important to reinforce that the aim of this work is not to develop models that accurately predict complications. Instead, it is to develop and validate a novel definition of complications, and use that to estimate rates of complications. Model performance is used to justify our novel definition as it identifies many more patients than using the OECD definition, where model performance is similar to the much more restricted OECD definition, and where outcomes in patients with complications are clearly worse than in those without complications. For that reason, we did not perform internal model validation (e.g., with a test-train-validate split, or K-fold validation) because the aim was to validate the list of complications, rather than the predictive model itself. Therefore, for the first time, we have a definition of post-operative complications in primary brain tumour patients that is assessable at a national level, using routine electronic health records. The model built using the extended list performs as well as one constructed using a much smaller list of OECD-defined complications. However, this does not mean that this model can easily be used to predict post-operative complications, although it may help inform such models. Equally, this is likely to be most applicable to England, although it may act as a basis for work in other countries. The aim of this work is to unlock the ability to count post-operative complications in brain tumour patients undergoing neurosurgery. This then allows us to explore predictors and consequences of such complications in a robust, large-scale, unbiased fashion at a national level, and may allow for use as a quality standard or for benchmarking between centres, for example.

Previous work has shown that surgeon volumes predict post-operative mortality in this group of patients. However, post-operative mortality is low and there is much less work on post-operative complications, in part because there are few large datasets to work with. This paper shows that we can measure post-operative complications in patients undergoing

neurosurgery for a primary brain tumour in England; that our list of complications captures many more patients than the OECD-defined PSI list, and yet maintains model performance. Future work will explore the mediators and implications of these complications in more detail. More information on the GlioCova project can be found here: https://blogs.imperial.ac.uk/gliocova/about-gliocova/.

## Supporting information

**S1 Appendix. Data and coding details.**
(DOCX)

**S2 Table. List of OPCS 4 codes used to define major resection and biopsy.**
(DOCX)

**S3 Table. OECD Patient safety (PS) questionnaire codes.** Codes were obtained from the OECD Health Care Quality and Outcomes (HCQO) 2018−19 Data Collection: https://www.oecd.org/statistics/data-collection/Health%20Care%20Quality%20Indicators_guidelines.pdf (* - codes excluded from our study).
(DOCX)

**S4 Table. The list of top 100 diagnosis codes extracted from index admission.**
(DOCX)

**S5 Fig. Flowchart for the modified Delphi process.**
(PDF)

**S6 Table. ECI comorbidity index variables and the corresponding ICD-10 codes used to identify them.**
(DOCX)

**S7 Table. Scores for modified Elixhauer Comorbidity Index variables using all brain/spinal tumour patient admissions between 2012–2019 based on English HES data.** A higher score indicates that this comorbidity is linked with higher log odds of in-hospital mortality (e.g., a score of 2 is associated with a double increase in the log odds of mortality).
(DOCX)

**S8 Fig. Flowchart for the selection of patients having major resection or biopsy.**
(DOCX)

**S9 Table. Distribution of the location of discharge for all patients undergoing first surgical intervention in the Gliocova dataset.**
(DOCX)

**S10 Table. Most common ICD-10 diagnoses that could indicate reason of readmission.**
(DOCX)

## Acknowledgments

This work uses data provided by patients and collected by the NHS as part of their care and support.

We want to thank Nicola Glover for collaboration regarding clinical coding.

## Author contributions

**Conceptualization:** Andrew Brodbelt, Kerlann Le Calvez, Thomas C Booth, Jonathan J Gregory, Matt Williams.

**Data curation:** Kerlann Le Calvez.

**Formal analysis:** Radvile Mauricaite.

**Investigation:** Radvile Mauricaite.

**Methodology:** Alex Bottle, Kerlann Le Calvez, Peter Treasure, Matt Williams.

**Supervision:** Kerlann Le Calvez, Matt Williams.

**Validation:** Andrew Brodbelt, Stephen J. Price, Seema Dadhania, Jonathan J Gregory, Maureen Dumba, Joanne Droney, Jawad Basharat.

**Writing – original draft:** Radvile Mauricaite, Thomas C Booth, Matt Williams.

**Writing – review & editing:** Radvile Mauricaite, Alex Bottle, Andrew Brodbelt, Kerlann Le Calvez, Peter Treasure, Stephen J. Price, Thomas C Booth, Jonathan J Gregory, Matt Williams.

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
