## [Decision Letter · Decision Letter 0]

16 Nov 2025

Dear Dr. Williams,

Thank you for submitting your manuscript to PLOS ONE. After careful consideration, we feel that it has merit but does not fully meet PLOS ONE’s publication criteria as it currently stands. Therefore, we invite you to submit a revised version of the manuscript that addresses the points raised during the review process.

**ACADEMIC EDITOR:**

We look forward to receiving your revised manuscript.

Kind regards,

Athanasios G. Pantelis

Academic Editor

PLOS ONE

Journal Requirements:

6. Thank you for stating the following in the Competing Interests section:

Dr. Williams is employed by Imperial College Healthcare NHS Trust. He is also the medical director of PearBio (salary and share options).

Stephen J. Price is an advisor for TUMOURVUE Ltd (no financial involvement) and is the Chair of the Education Committee of the European Association for Neuro-oncology (EANO). He is on the speaker board for Medac GmbH and organises courses to train surgeons to use 5-ALA for which he is reimbursed, but has not run such a course since 2019.

Alex Bottle declared obtaining fees from AstraZeneca and Lilly outside the submitted work.

7. Please amend your list of authors on the manuscript to ensure that each author is linked to an affiliation. Authors’ affiliations should reflect the institution where the work was done (if authors moved subsequently, you can also list the new affiliation stating “current affiliation:….” as necessary).

Reviewers' comments:

Reviewer's Responses to Questions

**Comments to the Author**

1. Is the manuscript technically sound, and do the data support the conclusions?

Reviewer #1: Yes

Reviewer #2: Yes

Reviewer #3: Yes

Reviewer #4: Yes

Reviewer #5: Yes

Reviewer #6: Yes

2. Has the statistical analysis been performed appropriately and rigorously?

Reviewer #1: Yes

Reviewer #2: Yes

Reviewer #3: I Don't Know

Reviewer #4: Yes

Reviewer #5: Yes

Reviewer #6: Yes

3. Have the authors made all data underlying the findings in their manuscript fully available?

Reviewer #1: Yes

Reviewer #2: Yes

Reviewer #3: Yes

Reviewer #4: Yes

Reviewer #5: Yes

Reviewer #6: Yes

4. Is the manuscript presented in an intelligible fashion and written in standard English?

Reviewer #1: Yes

Reviewer #2: Yes

Reviewer #3: Yes

Reviewer #4: Yes

Reviewer #5: Yes

Reviewer #6: Yes

Reviewer #1: This is an important and timely study that addresses a clear gap in neurosurgical outcomes research: the absence of brain tumour–specific, scalable definitions of post-operative complications. Using a large national dataset, the authors develop and validate a novel complication list (ICL list), showing it identifies more patients with adverse outcomes than the OECD/PSI definitions. The methodology is rigorous and transparent, and the findings are of international relevance.

Major comments

1. Clarity of aims and framing

- The introduction is comprehensive but could be streamlined. The central research question (“Can we use administrative data to define and validate a broader, brain tumour–specific complication list at national scale?”) is somewhat diluted. A sharper aim statement at the end of the introduction would strengthen the framing.

2. Definition and validation of the ICL list

- The Delphi process is described, but more detail is needed on how disagreements were resolved and how final codes were confirmed as complications. Were sensitivity analyses conducted excluding “borderline” diagnoses?

- It would be helpful to present the final ICL list (perhaps in the appendix) in a way that readers could apply in future work.

3. Statistical modelling

- The models are used only to validate the ICL list (not to predict outcomes). This is appropriate, but it should be more explicitly stated early to avoid reader confusion.

- The relatively modest C-statistics (<0.8) should be discussed further: what does this imply for future work on prognostication versus complication surveillance?

4. Interpretation of results

o The complication rate of 11.3% is convincing, but the clinical interpretation could be expanded. Which complications are driving this increase compared to OECD? Are they mainly infections, neurological events, systemic issues? This breakdown would aid clinical applicability.

- The apparent paradox that meningioma patients have higher complication rates but lower mortality than GBM patients deserves more discussion.

5. Implications for practice and policy

- The discussion could expand on how this work might be used in practice: e.g., benchmarking between centres, informing quality improvement, linking complication burden with health economic analyses.

- The authors should comment on whether the ICL list might be adaptable beyond England, or whether country-specific coding systems would require recalibration.

Minor comments

- At times “OECD list,” “PSI,” and “OECD-defined complications” are used interchangeably; consistent language would improve clarity.

- ROC curves and cohort flowcharts are central to the argument — ensure they are high quality and easy to interpret.

- A few citations (e.g. [6], [39–41]) could be updated or clarified to ensure they align with the statements in text.

- Tighten the introduction to avoid repetition, and consider shortening the methods for clarity (details can be shifted to supplementary).

Recommendation

The study is methodologically strong and of high clinical importance. Clarifying the aims, providing more transparency on the ICL list development, expanding the interpretation of findings, and sharpening the framing of implications will substantially strengthen the manuscript and broaden its impact.

Reviewer #2: This is a well-written and thoughtfully conducted study that provides new insight into postoperative complications following primary brain tumor surgery. By establishing a more comprehensive and specific list of complications, the authors enhance our ability to estimate and understand the true burden of postoperative morbidity in this patient group. The study contributes meaningfully to improving ethical and balanced discussions with patients and families, supporting more informed decision-making and risk communication.

The analysis adds significant value to national data interpretation and future policy development. However, the study could be further strengthened by addressing how such findings might guide therapeutic decision-making or influence perioperative management strategies to mitigate risk. Overall, it represents an important step toward refining outcome measurement and advancing patient-centered neurosurgical care.

Reviewer #3: PONE-D-25-41552

Estimating post-operative complication rates in patients with primary brain tumours from routine administrative data: A National Cohort study

In this cohort study the authors analysed national administrative data from 29,018 adults undergoing brain tumour surgery in England to develop and validate an expanded definition of postoperative complications (the ICL list). They found that this definition identifies over five times more complications than current OECD indicators and conclude it enables meaningful estimation of neurosurgical complication rates at a national level.

This is a very timely and well-designed study that addresses an important gap in neurosurgical outcomes research. The development of a scalable definition of postoperative complications using national administrative data is innovative and highly relevant for both clinical practice and health service evaluation.

Comments

The abstract for this kind of study should be presented in a structured format (Introduction, methods etc…)

The introduction should be reworked: it is redundant and could be shortened, the aim should also be reported more clearly.

The development of the ICL list is not described with sufficient clarity. The Authors should provide clearer criteria for how diagnoses were selected as complications and briefly explain the Delphi process used, including how consensus was reached.

The definition of the ICL complication list is central to the manuscript, but the full content of the ICL list only appears in supplementary material. The Authors should include either the full list or at a clearly structured summary table in the main text.

The section titles that divide the discussion are not necessary and could be removed.

The Authors should better address the generalisability of their findings. Although the cohort is nationally representative within England, it is unclear how well the ICL complication list would apply to other healthcare systems with different coding practices or neurosurgical pathways. A brief discussion of this limitation is recommended.

The conclusions section does not clearly summarise the main findings of the study. Instead, it focuses on future work without explicitly stating the key result: the ICL list identifies more postoperative complications than existing OECD indicators while maintaining model performance. The Authors should revise the conclusion to clearly state the study’s primary outcome. Phrases like “(by us and many others)” should also be avoided.

Reviewer #4: This is a well-executed national study addressing an important estimation of complication rates after brain tumour surgery using administrative data. The methodology, dataset, and statistical analysis are sound, and the manuscript is generally clear and well organized. The study’s use of the new Imperial College London (ICL) complication list is a meaningful advance, as it captures neurosurgery-specific complications missed by standard lists like the OECD’s. To improve clarity, the authors should better highlight what is novel about the ICL list in the introduction, summarizing how it was derived and validated. The methods section would benefit from a brief description of the main Delphi criteria used to select complications. The discussion should briefly address how this framework could be applied within NHS quality programs and its potential use in benchmarking or risk adjustment. Minor edits include shortening the abstract and defining “ICL list” at first mention, clarifying whether internal validation methods were used, and checking for minor errors. Overall, this is a strong and valuable contribution that merits publication after minor revisions for clarity and emphasis on the study’s novel aspects.

Reviewer #5: Summary

In this paper the authors analyzed post-operative complications of adult brain tumor surgery on a large scale. They performed a retrospective cohort review of a national brain cancer database (GlioCova) in England between 2013-2018. The goal was to assess the rate of complications without individual chart review. First, they defined a novel list of brain tumor surgery complications (so-called ICL-list) with 51 diagnoses as coded by the ICD-10. They defined surrogate parameters to measure the rate of post-operative complications such as 30-day mortality (30-DM) and length of stay (LOS). They validated their ICL-list by running statistical models and comparing the performance to the list of Patient Safety Indicators (PSI) as developed by the US Agency for Health-care Research and Quality (AHRQ). As opposed to the novel ICL-list the PSI-list assess patient safety and is only defined by a very small number of key diagnoses. As a result, the authors analy-zed a cohort of 29,018 patients. The cohort was subdivided into age groups (18-40, 40-65 and 65+) and tumor type (glioblastoma / meningioma). They showed that the PSI-list captured a smaller proportion of patients with complications compared to the newly defined ICL-list and that both performed well in predicting 30-day mortality and length of stay. The novelty of their approach was to be able to use large scale healthcare data, rather than manually retrieved clinical chart data. The complication rate was 11.3%, the 30-day mortality was 2.3% and length of stay 5d and the 30-day readmission rate 12.7%. Older patients or with more co-morbidities, were more likely to have complications or die. They thoroughly discussed their findings in the light of the existing literature. In the future they plan to analyze the impact of complications on costs and adjuvant treatments.

Comments

The authors are to be commended! To our knowledge their work is a novel approach to analyze complications of brain tumor surgery on such a large scale and to be able to extract data from routine health care databases. Interestingly, the extended ICL-list performs similar to the more restricted PSI-list in statistic models. The authors may further comment on this in the discussion part.

The manuscript is clearly written and needs no further language editing.

Please, briefly define and describe the ICL-list (novelty/ difference) in the abstract section.

Typos

p. 2, l. 47 define “ ICL list”,

p. 4, l. 101: typo “… be longer…”,

p 4., l.106: typo “… them. In this work…”

Reviewer #6: I have read this manuscript with great interest. The authors address an important gap in neurosurgical quality assessment by developing a comprehensive complication list measurable at national level using administrative data. The study leverages an impressive dataset of nearly 30,000 patients with appropriate statistical methods. The manuscript is well-written and the research question clinically relevant. However, I have several concerns that need to be addressed before publication.

Major Concerns

1. Validation of the ICL Complication List

The validation strategy is insufficiently detailed. The Delphi process (S5 Figure) lacks critical information: What criteria did the five clinicians use to classify diagnoses as complications versus pre-existing conditions? How was consensus defined? What was inter-rater agreement?

The validation against a "local dataset with access to detailed clinical notes" (lines 155-156) is too vague. How large was this dataset? How many of the 11 GlioCova-defined complications were verified as true complications? What proportion of coded diagnoses were confirmed when clinical notes were reviewed? Without this information, readers cannot assess whether the ICL list truly captures complications or includes pre-existing conditions. The statistical validation—showing ICL models perform similarly to OECD models—is clever but indirect. It demonstrates that the expanded list includes diagnoses associated with worse outcomes, but does not prove all 51 diagnoses represent true complications. The authors should provide detailed validation data including inter-rater agreement from the Delphi process, and quantitative validation results from the local dataset, even if only for a subset of complications.

2. Coding Accuracy and Data Quality

The limitations of administrative data are inadequately addressed. The authors mention that "some diagnoses might be miscoded or missed" (lines 331-332) without quantitative assessment. Several issues need attention:

First, inter-center variability in coding practices is not examined, though this could lead to spurious variation in complication rates. Second, there is no discussion of financial incentives in the English NHS system—do hospitals receive additional payment for treating complications? Third, implications of coding accuracy (cited as ~80% in UK literature) for the estimated 11.3% complication rate are not discussed.

Please add sensitivity analysis stratifying complication rates by center or region, if possible. Discuss financial coding incentives and their potential impact. Add a paragraph on how coding accuracy might affect findings, including direction and magnitude of potential bias.

3. Temporal Assignment of Complications

How was it ensured that coded diagnoses represent post-operative complications rather than pre-existing conditions? ICD-10 coding in England does not consistently mark "present on admission." The authors excluded "chronic illnesses" (line 146), but many conditions could be either pre-existing or post-operative. For instance, headache is listed among common readmission diagnoses (S10 Table), but is also a common presenting symptom of brain tumors. This is particularly relevant for the 90.3% of complications from the GlioCova-defined list (line 216), which are not shown in the main manuscript.

Please explicitly describe the strategy for ensuring diagnoses represent post-operative complications. Consider sensitivity analysis using only unambiguously post-operative complications (e.g., surgical site infection, post-operative hematoma). Present the 11 GlioCova-defined complications in the main text, not just in supplement.

Minor Concerns

4. Statistical Issues

The dichotomization of LOS at 3 days (line 170) appears arbitrary—was this clinically or data-driven? The C-statistics are moderate (0.77 for mortality, 0.66 for LOS), limiting the strength of the validation approach. There is no mention of how missing data were handled.

Justify the 3-day LOS cutoff. Clarify missing data handling.

5. Clinical Interpretation

Table 2 shows a paradox: meningioma patients have higher complication rates than glioblastoma patients (e.g., 13.6% vs. 8.9% in age 18-40), yet much lower 30-day mortality (0.5% vs. 5.1%). Are meningioma complications less severe? Is this due to different surgical approaches or coding practices? This warrants discussion.

Additionally, 30-day readmission is presented prominently but not used as an outcome in validation models. Why not? Discuss the meningioma-glioblastoma paradox. Consider including readmission in validation analyses.

6. Presentation Issues

For a methods paper defining a new complication list, relegating the complete ICL list to supplement seems inappropriate. The ethics statement "N/A" is inadequate—at minimum, describe the data governance structure and institutional approval. Data availability states researchers must "contact us to enquire," but more clarity on the access process would aid reproducibility. Move at least the 11 GlioCova-defined complications to the main text. Provide proper ethics statement with data governance details. Clarify data access procedures.

Minor Editorial Points

Line 106: "we just want to be able to count them"—too colloquial, rephrase formally

Line 200/Table 1: Age quintile notation confusing (1st-18, 2nd-46)—should be Q1=18, median=58, Q3=68

Table 1: "C71 Other" (16.5%) is very heterogeneous—provide more detail

Lines 222-225: LOS data with quintile notation is unreadable—use a table

**Do you want your identity to be public for this peer review?** For information about this choice, including consent withdrawal, please see our Privacy Policy

Reviewer #1: No

Reviewer #2: No

Reviewer #3: No

Reviewer #4: No

Reviewer #5: No

Reviewer #6: No

---

## [Author Response · Author response to Decision Letter 1]

14 Jan 2026

Please see the 'Response to reviewers' document attached to the submission. We believe we have answered all of these in the document.

---

## [Editor Report · Decision Letter 1]

15 Jan 2026

Estimating post-operative complication rates in patients with primary brain tumours from routine administrative data: A National Cohort study

PONE-D-25-41552R1

Dear Dr. Williams,

We’re pleased to inform you that your manuscript has been judged scientifically suitable for publication and will be formally accepted for publication once it meets all outstanding technical requirements.

Kind regards,

Athanasios G. Pantelis

Academic Editor

PLOS One

Additional Editor Comments (optional):

All reviewers’ comments have been adequately addressed by the authors, and the manuscript is compliant with the relevant EQUATOR reporting guideline (STROBE).
---

## [Editor Report · Acceptance letter]

PONE-D-25-41552R1

PLOS One

Dear Dr. Williams,

I'm pleased to inform you that your manuscript has been deemed suitable for publication in PLOS One. Congratulations! Your manuscript is now being handed over to our production team.

Kind regards,

on behalf of

Dr. Athanasios G. Pantelis

Academic Editor

PLOS One